# Patient Perspectives on Social and Identity Factors Affecting Multiple Myeloma Care: Barriers and Opportunities

**DOI:** 10.3390/healthcare12161587

**Published:** 2024-08-09

**Authors:** Natalia Neparidze, Amandeep Godara, Dee Lin, Hoa H. Le, Karen Fixler, Lisa Shea, Stephanie Everson, Christine Brittle, Kimberly D. Brunisholz

**Affiliations:** 1Yale School of Medicine, Yale University, New Haven, CT 06510, USA; natalia.neparidze@yale.edu; 2Huntsman Cancer Institute, Salt Lake City, UT 84112, USA; amandeep.godara@hci.utah.edu; 3Janssen Scientific Affairs, LLC., Horsham, PA 19044, USA; dlin@its.jnj.com (D.L.); hle15@its.jnj.com (H.H.L.); kfixler@its.jnj.com (K.F.); lshea4@its.jnj.com (L.S.); 4Independent Researchers, Goodyear, AZ 85395, USA; misseverson@yahoo.com; 5CorEvitas, Waltham, MA 02451, USA; cbrittle@corevitas.com

**Keywords:** multiple myeloma, patient-centered care, health equity, access to care, perceived social supports

## Abstract

Patients living with multiple myeloma (MM) have a substantial disease burden and face multiple barriers to care. Building upon our previous research using mixed methods, this focus group research aimed to identify patients’ priorities regarding specific social and identity-related needs, map these prioritized needs to the disease journey, and describe patient-generated ideas to improve patient support. Participants noted that patients with MM need a range of emotional, social, and financial support throughout the disease journey. They identified initial MM diagnosis and treatment adherence as two critical points in the MM journey where patients need the most support and assistance. The findings of this research suggest that overall, patients with MM need comprehensive support, ideally from a multidisciplinary team consisting of health care providers, patient advocates, social workers, and psychologists to help patients understand their disease and treatment options, make informed treatment decisions, adhere to treatment, and ultimately reduce their disease burden and improve outcomes. This research revealed that patients with MM need varying types and levels of support, with the most common needs including information on disease and treatment, connections to financial resources and support systems, assistance with navigating insurance options, and transportation and logistical support for medical appointments.

## 1. Introduction

Multiple myeloma (MM) is a heterogeneous hematologic malignancy affecting plasma cells in the bone marrow. It is the second most common blood cancer in the United States (US) and is commonly diagnosed in older adults (median age at diagnosis of 69 years) [1]. Compared with non-Hispanic White patients, Black and Hispanic patients with MM tend to be younger at initial diagnosis [2,3], are less likely to undergo needed diagnostic testing [4], have limited access to novel treatment options [5,6], experience more treatment interruptions [5,6], and receive less supportive care [7,8].

Overall survival of MM has drastically improved in the past two decades because of the introduction of novel targeted and immunotherapies, including chimeric antigen receptor T-cell therapies and bispecific antibodies. Despite advances in treatment, a complete cure for MM remains difficult to achieve. Almost all patients with MM eventually experience relapse and become refractory to treatment, and most patients require long-term treatment during their lifetime [9,10]. As a result, MM has transformed into a chronic disease for many patients.

As patients navigate the long disease journey, understanding their perspectives on disease burden, barriers to care, and support needed is becoming increasingly important to optimize patient care. In our previous research using mixed methods, 65% of patients reported ≥1 social need(s) that negatively impacted health outcomes including lack of knowledge about MM, financial instability and lack of insurance, transportation concerns, and lack of social support [11]. Learning directly from patients where prevalent social needs and experiences occur on the MM diagnostic and treatment journey is a critical gap that needs to be investigated in order to effectively design mitigation solutions and alleviate MM disease burdens.

To build upon the findings of our previous research, we designed a series of exploratory focus groups to identify patients’ priorities with respect to their specific social and identity-related needs. We aimed to map these prioritized needs and experiences to the MM diagnostic and treatment journey and describe patient-generated ideas to improve the support that patients need. Herein, we report the findings of this follow-up research.

## 2. Materials and Methods

### 2.1. Study Design

This qualitative study consisted of 2 sets of 4 semi-structured focus group discussions (8 groups in total) in which we built our research questions upon findings from our previous research [11]. The first set of groups focused on understanding patients’ experiences with MM and identifying where along their journeys these factors have the most impact. The second set of groups focused on patient-generated ideas to improve needed support.

### 2.2. Participants

Participants were adult (≥18 years of age) US residents with MM who were currently in their first or second line of therapy and were existing members of the sponsor’s Multiple Myeloma Patient Engagement Research Council (PERC), which has been described elsewhere [12]. Patients ≤ 18 years old and those with receipt of stem cell transplant > 5 years from the time of PERC initiation were excluded. In summary, PERCs include a diverse group of patients with chronic diseases and varied healthcare experiences who voluntarily participate in specific research activities.

At the time of this study, the MM PERC consisted of 23 patients living with MM, who had all received some treatment, including stem cell transplant. Participants were informed that no treatments would be provided, and they could withdraw at any time. Additionally, a consent and release form was signed by the participants that communicated confidentiality and Health Insurance Portability and Accountability Act (HIPAA)-compliant practices. All data were deidentified and collected for improvement purposes, and an ethics board deemed the study exempt from IRB review according to the Department of Health and Human Services Policy for Protection of Human Research Subjects (Section 45C.F.R46.104). The study was also conducted in accordance with the Helsinki Declaration of 1964 and its later amendments. All sessions were conducted virtually, and participants joined from their homes or another location [12].

### 2.3. Data Collection

All MM PERC members were invited to participate in 2 sets of online focus group sessions conducted in September and November 2023. All focus group sessions were semi-structured and led by an experienced qualitative researcher (CB) who followed a discussion guide (Appendix A). Each session included 4 to 6 PERC members and lasted 2 h. Focus group discussions were audio-recorded and transcribed verbatim.

### 2.4. Data Analysis

At the conclusion of each set of focus groups, the research team (LS, KF, KB, HL, DL, and CB) met to debrief and identify initial themes. Transcripts were analyzed using narrative thematic analysis to identify and organize themes according to key research questions related to patient experiences, how social needs and identity experiences map to the patient journey, and key ideas for patient support. Data were reviewed in totality and findings were assessed thematically using qualitative coding to ensure rigor based on the key questions in the discussion guides (Appendix A).

## 3. Results

### 3.1. Participant Characteristics

A total of 23 MM PERC members participated in this research, including 17 members who participated in both sets of focus groups, 2 in the first set only, and 4 in the second set only. Participant sociodemographics are summarized in Table 1. Most participants were female, less than half were White, more than half were in their 40s or 50s, and the majority had a bachelor’s or graduate degree (Table 1).

### 3.2. Describing Key Characteristics of Living with MM

Participants elaborated on the three key characteristics of life with MM identified in our previous research [11], including ongoing anxiety, frequent side effects, and limited knowledge about MM at initial diagnosis. Participants agreed that there is a significant mental health burden associated with MM and reported that their ongoing anxiety was driven by multiple factors. These factors include fear of relapse (particularly heightened while waiting for the results of ongoing monitoring tests), treatment failure (common for MM), the ongoing need to decide on new courses of treatment, the significant out-of-pocket expenses associated with MM treatment and the related loss of income for many who were no longer able to work, and uncertainty about future care.


*“I have to have testing, the MRIs, and certain things, bone marrow biopsies once a year, and so forth. And that can lead to quite a bit of anxiety… And then after the scan, you’ve got to wait sometimes a week or more to get the results.”*
(White patient, 50s)


*“Most of the treatments that you get have a limitation of 18 months to maybe 2 years. So, you’re kind of, like, on pins and needles. Is this thing going to take me all the way out? …What is the next on the horizon? … Is it going to come back harder than it was before?”*
(Black patient, 50s)

Most participants also affirmed that MM is associated with frequent treatment side effects; however, a few participants shared that they had limited or no side effects. Physical fatigue was a frequently reported side effect, which disrupted daily activities, reduced quality of life, created challenges in managing other concurrent health conditions, and required patients to manage their energy levels. Side effects were also noted as somewhat unpredictable, which contributed to mental fatigue and anxiety about whether their MM was progressing. Participants shared that side effects could weaken their immune system, making them more vulnerable to other illnesses.


*“The longer I’m on a particular medication…new side effects crop up. And so, I’m always managing something. When you think you’ve sort of mastered it or have it managed, something new springs up. It’s just endless.”*
(White patient, 60s)


*“Not only physical fatigue, [but it’s also] mental fatigue. … The only way I can describe it is indifference. You don’t care about anything.”*
(Hispanic patient, 40s)

Most patients shared that they knew little about MM at the time of diagnosis; the only exceptions were patients who personally knew someone with MM prior to their own diagnosis or who had access to an MM specialist right away. Limited knowledge contributed to additional challenges such as delayed access to specialized care, lack of understanding of available treatment options (particularly advanced treatment options), difficulty figuring out insurance coverage, and heightened anxiety due to concerns about long-term prognosis.


*“I didn’t have a clue of what multiple myeloma was when I was diagnosed with it.”*
(Asian patient, 70s)


*“I didn’t have any information when I was told that I had multiple myeloma. I was diagnosed with it in 2015. I didn’t know anybody else that had it. I didn’t know anything about it. So, yeah, the information was very small.”*
(Black patient, 60s)

### 3.3. Comparing MM to Other Chronic Health Conditions

Our previous research identified six main areas in which MM is different from other chronic health conditions [11]. Participants in this study further explained the key differences in these areas (Table 2). They noted that diagnosis of MM could be delayed, especially among younger patients, because of patients’ limited knowledge about the signs and symptoms of MM and a misconception that younger people do not develop MM. Participants pointed out that MM progression is uncertain and varies from patient to patient, and that almost all patients have to deal with relapses, which lead to long-term treatment. Participants noted that the financial burden associated with MM is substantial largely because of the high costs of advanced treatment and supportive care. Participants emphasized that MM is associated with significant physical and mental health burdens, which could lead to disability and unemployment, and worsening financial concerns. Additionally, participants noted that the continuous treatment and periodic testing associated with MM management require frequent medical appointments and may involve lengthy hospital stays.

### 3.4. Social Needs and Identity Experiences Mapped to Patient Journey

Our previous research identified discrete social needs and identity experiences that can negatively impact the patient’s journey. In this current study, participants were asked to map these discrete factors to the patient journey and to identify at which of the six stages during the journey (diagnosis, treatment goals, treatment choice, treatment initiation, treatment adherence, and outcome/disease burden) these factors have the greatest impact.

Participants’ mapping of these prioritized social needs and identity-related experiences is summarized in Figure 1. Full patient responses are summarized in Table 3. Subsequent results are organized by disease journey where patients describe the critical social needs at each of the six stages.

#### 3.4.1. Diagnosis

Participants identified two major factors negatively impacting diagnosis: lack of health knowledge and cultural background. For some participants, a lack of knowledge about MM contributed to delays in diagnosis. For example, one participant reported visiting primary care five times complaining about being unwell, but she did not self-advocate for diagnostic testing, such as for an MRI, until her leg pain worsened to the point that she could no longer stand. Some participants reported that their culture encourages minimizing pain and/or discourages patients from seeking treatment, and some also shared that their families encouraged them to “push through” their initial symptoms. As a result, these patients had advanced disease at the time of diagnosis.


*“In the Latino culture, everything is about work and to push through things and be strong. And so that was the message I was getting from my dad, from my husband, from my family. I didn’t feel well. … ‘You’ve just got to keep going. You’ve got to keep pushing.’ And I was frigging sick.”*
(Hispanic patient, 40s)

#### 3.4.2. Treatment Goals

Participants noted that treatment goals can be negatively impacted by a lack of social support, age, and disability status. Some participants shared that they needed an opportunity to discuss treatment options with and needed encouragement from others when determining treatment goals. Participants agreed that treatment goals are generally related to a patient’s age. One participant shared that her father, who also had MM, was not given many treatment options because of his age and died 2 years after diagnosis. Participants noted that disability also affects treatment goals because disability reduces patients’ ability to work. Participants who were no longer able to work because of their MM shared that they faced additional financial obstacles to treatment, which in turn affected their treatment goals.

#### 3.4.3. Treatment Choices

Of the six patient journey stages discussed in the focus groups, treatment choices were seen as most impacted by numerous social needs and identity experiences. Participants reported that treatment choices can be negatively impacted by a wide variety of factors, including health knowledge, financial instability, lack of insurance, lack of transportation, race and ethnicity, age, and disability status.

Participants noted that lack of health knowledge can negatively impact treatment choice because patients may be unaware of all treatment options, which is especially common when patients have delayed access to a specialist.


*“[I] was told there’s nothing I could do.…And I was unaware of all the different…guidelines and all these legal policy guidelines that dictated my choices, supposedly.…So, I had to find out for myself and self-educate and self-advocate.”*
(Asian patient, 50s)

Participants reported that financial instability and insurance coverage also limit treatment choices. Several participants shared that they had to stop working because of disability, which limited their financial resources and affected their insurance coverage. One participant shared that she reached her USD 5000 annual copay limit in January and struggled to find resources to pay the entire amount at the start of the year. Another participant shared that she opted to drive 2 h to receive an injection (instead of self-injecting at home) because that was a more affordable option based on insurance coverage. Another participant shared that their insurance would not cover access to specialists.

Participants reported that lack of transportation can also impact treatment choice. Some participants shared that some of their specialists were more than an hour away, making it hard for them to access specialized care.


*“Everybody said, ‘You need a specialist.’ But it’s just too far away to drive.”*
(Black patient, 60s)

Two other participants added that they were not able to participate in a clinical trial because of lack of transportation.

Participants indicated that race, ethnicity, and age can also impact treatment choice. One participant pointed out that there are fewer stem cell donors who are not White, which made it challenging for her to find a match. She shared that “*it was very hard for me to find a donor [because of my race]. So, I wind up having to go do a half match with one of my children*” (Black female, 50s). Participants reported that age significantly impacted treatment decisions and choices.


*“When I was first diagnosed, my oncologist…talked to me about a stem cell transplant…and I was 79 at the time. He said, ‘We don’t usually recommend it for people your age.’”*
(White patient, 80s)

Some participants reported that disability status can impact treatment options because disability makes it difficult to endure the frequent testing and visits associated with MM treatment.

#### 3.4.4. Treatment Initiation

Participants noted that lack of insurance coverage and issues with insurance can negatively impact treatment initiation. While all participants in this research had health insurance they reported experiencing treatment delays due to wait times for approvals or reauthorizations from their insurance providers as well as delays in obtaining access to specialized care. One patient shared that she was waiting for the next open enrollment period to switch to an insurance plan that would allow her to see a preferred specialist because *“my insurance does not cover this doctor. … So now I have to wait and check and see what other plans are available”* (Hispanic patient, 40s).

#### 3.4.5. Treatment Adherence

Participants reported that lack of social support and transportation as well as disability status can negatively impact treatment adherence. Participants shared examples related to how lack of social support had impacted their adherence. One participant reported missing appointments because her employer was not supportive, and she was afraid to ask for time off from work. Another participant shared that she ended up leaving a job because of an unsupportive employer, and a third participant shared that she was not able to receive a stem cell transplant because she did not have adequate social support to help care for her after the procedure.

Participants also shared that lack of transportation can impact treatment adherence. One participant shared that she was initially in a wheelchair and was unable to drive to appointments. Some participants were either nondrivers or were not able to drive following certain treatments. One participant shared that she had to “*Band-Aid together*” ride options for her care. Another participant moved his entire family to be closer to his specialist. Participants noted that even when transportation assistance was available to them, they still faced challenges because transportation assistance often had to be scheduled weeks or months in advance.


*“I was always trying to find transportation. And there was public transportation available…but they wanted you to sign up 2, 3 months in advance. And sometimes…the doctor all of a sudden wanted you to come in.”*
(White patient, 50s)

Participants reported that disability is another factor that can impact treatment adherence. One participant shared that sometimes he struggled to attend needed medical appointments because of his weakened physical condition. “*There were times after I went through treatment where, just the physical part, going to appointments and going to treatment and all that was difficult because I was so weakened*” (Hispanic patient, 60s).

#### 3.4.6. Burden of Disease and Disease Outcomes

Participants viewed disability as the primary factor negatively impacting disease burden, which could worsen over time. Several participants shared that they had to retire because of the disability caused by MM.


*“I am disability retired. Being a teacher, I kind of pushed myself to continue to teach. And that was a negative, because I wound up making myself sicker.”*
(Black patient, 50s)

#### 3.4.7. Key Parts of Patient Journey Where Support Is Most Needed

When asked at which point during their MM journey support would be most needed, participants noted that support is needed throughout the disease journey and especially at the initial diagnosis and treatment adherence stages. When asked to allocate 100 points among the six stages of the patient journey (Figure 1), giving more points to stages where they felt more support was needed, participants gave the diagnosis stage the most points (average 41 points), followed by treatment adherence (average 22 points).

Participants further shared that the initial MM diagnosis was a scary, overwhelming, and traumatizing time, and they needed support to understand available treatment options and make informed treatment decisions. Participants viewed the diagnosis stage as a critical time when they needed to quickly connect with supportive resources and establish support for the treatment journey.


*“It starts with the diagnosis. Once you’ve been diagnosed, putting you into the right environment for you to be successful and for you to be stable, and not necessarily lose your mind. … And if you don’t have the support that you need in order to, kind of, get through the initial phases, …you can’t even think about what the treatment options are.”*
(Black patient, 50s)

While participants also viewed social support as critical for treatment adherence, the type and level of support needed varied substantially from participant to participant, reflecting the broad needs that participants identified.


*“For some people, depending on their age, it’s a work barrier. For some people, it’s financial barrier. For some people, it’s a knowledge barrier. So, you’ve got to figure out which one of those pods can you help expand some people’s ability to be able to adhere to their treatments.”*
(Black patient, 50s)

### 3.5. Patient-Generated Ideas to Improve Support

In the second set of focus groups, participants generated ideas for improving the support they needed. They provided overall recommendations and suggestions specific to diagnosis and treatment adherence.

#### 3.5.1. Overall Recommendations

Participants indicated that patients with MM need comprehensive, personalized support, including the ability to meet frequently with a team who can help navigate the challenges of living with MM. They shared that ideally, support should be available throughout the entire journey, starting immediately after diagnosis, and the support team should include healthcare providers, patient advocates, social workers, and psychologists. Additionally, participants noted that they would benefit from hearing from other patients with MM about how they navigate the disease journey and receiving tips, support, and encouragement.

#### 3.5.2. Diagnosis-Specific Recommendations

Participants offered the following thoughts and suggestions for improving support at the initial MM diagnosis stage:Training for providers may be helpful in delivering the diagnosis more effectively and compassionately and in connecting patients with supportive resources.Patients with MM need immediate support to help process their diagnosis. Patients also need sufficient time to absorb the diagnosis because they may not have the emotional capacity or prior knowledge to fully absorb the information immediately.Patients need basic information about MM, ideally in their native languages, to help them understand their diagnosis and be reassured that a long life is possible. Early information should include the availability of MM specialists (participants who had access to a specialist earlier in their journey were more satisfied with their initial treatment). Patients also need to learn how to read lab results and be referred to reliable sources for additional information. Subsequent information should explain more detailed treatment options, including advanced treatment options.Patients need support to navigate and understand their insurance options, find financial assistance opportunities, such as copay support or foundations that provide financial assistance, and learn about the availability of transportation or other logistical support.

#### 3.5.3. Adherence-Specific Recommendations

Participants suggested a range of options to help improve treatment adherence, reflecting their varied needs. They noted that the following are important for adhering to treatment:Available information on treatment side effects and continued emotional support to deal with the side effects.Connections to financial resources to overcome the financial barriers associated with treatment costs and help navigate insurance.Transportation and logistical support to get to appointments, ideally provided without any out-of-pocket costs because finances may be an issue for some patients.Information patients can provide to employers on the length and frequency of required medical visits as well as how to provide appropriate accommodations for patients with MM who remain in the workforce.Tools or reminders to help patients adhere to medication schedules (e.g., calendars, alarms, organizers, notes, and apps).Tips from other patients to improve adherence, such as utilizing third-party mail services to receive shipments of medications so that patients do not have to be home to receive them.

## 4. Discussion

Consistent with previous studies [13,14], we found that patients with MM endure substantial mental and physical health burdens, including ongoing anxiety and fatigue due to ongoing MM treatment and disease monitoring. This research revealed that patients with MM need a wide range of supports throughout the disease journey to help them process initial diagnosis, understand treatment options, and improve treatment adherence while navigating the social and financial implications of MM and reducing disease burden. Patients with MM who participated in this research identified a list of specific social needs and barriers that are most impactful to them and suggested ideas to meet these needs. As patient perspectives become increasingly important in MM treatment decision-making [15,16], these patient-generated ideas offer valuable insights and can be leveraged to create concrete solutions to overcome the barriers and improve the support that patients need.

Patients in this research identified the initial MM diagnosis and treatment adherence as two critical points during the MM journey where they need the most support and assistance. The most common needs include disease- and treatment-related information, social and emotional support, financial assistance, and logistical support (such as transportation). Consistent with the findings of our previous research [11], lack of understanding of MM, especially at initial diagnosis, and the pros and cons of different treatment options are knowledge gaps, underscoring the need to provide patient-facing educational materials, ideally in multiple languages, to help patients understand the disease and make informed treatment decisions. Because patients’ knowledge of MM and information needs evolve over time, different educational materials are needed to support patients throughout the disease journey, including high-level, easy-to-understand information about MM and available treatment options at initial diagnosis, and more specific information (e.g., comparisons between treatment options, what side effects to expect and how to manage them, and insurance coverage, etc.) as patients go through cycles of treatment.

This research also indicated that social and emotional support play critical roles in MM management. Because MM has a substantial impact on the patient’s mental health, many patients will benefit from connections with peers who might be able to offer relatable support. Additionally, participants suggested the need for healthcare providers to improve their communication skills which would ultimately lead to deeper, authentic provider/patient relationships, especially when delivering the initial diagnosis.

Given the heterogeneous nature of MM and the varied socioeconomic statuses of patients with MM, it is no surprise that this research found that patients need various types and levels of support. While most patients need support to navigate the healthcare system and understand their insurance coverage, many may also need help to find financial assistance to offset treatment costs. Additionally, some patients may also need help to find available transportation programs that they can use for their medical appointments.

To our knowledge, this is the first research to map patients’ social needs and identity experiences to the MM journey and describe patient-generated ideas to overcome barriers and improve patient care. Our findings clearly show that it is paramount to connect patients with a wide range of supportive resources upon initial MM diagnosis and throughout the disease journey. Nonetheless, this study has a few limitations. First, this exploratory study included a modest number of patients, which may limit the generalizability of the findings. Second, all participants in this study were members of the sponsor’s PERC program and had health insurance. Compared with the general population of patients with MM, these patients have a higher level of education and may be more engaged in their treatment decision-making. Third, while participants were all diagnosed with MM and have received treatment, variation in treatment duration, treatment regimen types and disease phase is expected given the heterogeneity and treatment complexity of the disease.

## 5. Conclusions

In this study, participants prioritized critical social needs delineated by the disease journey stage and presented ideas that could be used to improve patient care. The findings of this study will help patient navigators, advocacy groups, community care organizations, as well as health care professionals better support patients with MM as they go through their unique and challenging disease journeys.

## Figures and Tables

**Figure 1 healthcare-12-01587-f001:**
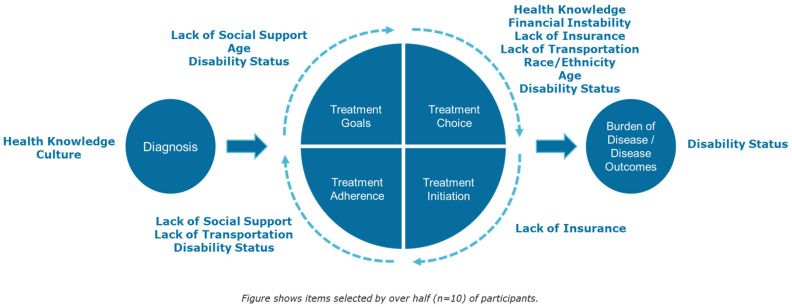
Participants identified critical social needs and identity experiences by stage of patient journey.

**Table 1 healthcare-12-01587-t001:** Demographic characteristics of MM focus group participants.

Category	First Set of Groups (*n* = 19) *n* (%)	Second Set of Groups (*n* = 21) *n* (%)
** *Gender* **		
Female	12 (63.2)	14 (66.7)
Male	7 (36.8)	7 (33.3)
** *Race and Ethnicity* **		
White	8 (42.1)	9 (42.9)
Black	5 (26.3)	6 (28.6)
Asian	2 (10.5)	2 (9.5)
Hispanic	4 (21.1)	4 (19.0)
** *Age Range* **		
40s	3 (15.8)	3 (14.3)
50s	9 (47.4)	9 (42.9)
60s	5 (26.3)	6 (28.6)
≥70s	2 (10.5)	3 (14.3)
** *Educational Achievement* **		
Graduate degree	8 (42.1)	8 (38.1)
Bachelor’s degree	4 (21.1)	4 (19.0)
Some college	5 (26.3)	7 (33.3)
High school or less	2 (10.5)	2 (9.5)

**Table 2 healthcare-12-01587-t002:** Patient perspectives on how MM varies from other chronic health conditions (*n* = 19).

Perceived Areas of Difference	Supporting Quotation
** *Diagnosis can be delayed* **	
Overall awareness of MM is low, contributing to delayed diagnoses. Diagnosis of MM could be particularly delayed in younger patients, and late diagnosis was seen as contributing to more advanced disease.	*“It seems like most people are diagnosed with this disease when they start having serious bone issues, and by that time, it’s already spread to a pretty serious level.” (White patient, 50s)*
** *Disease progression is uncertain and individualized* **	
There is a great deal of uncertainty related to when and how MM will progress. Patients noted the disease can progress differently between patients.	*“When you’re in remission, I don’t know if that’s worse, because you’re like, ‘OK, is it going to be tomorrow? Is it going to be the next week? Can I make plans?’ Because your plans are subject to change. It’s difficult.”* *(Hispanic patient, 60s)*
** *Treatment is ongoing and side effects can be significant* **	
Treatment for MM is ongoing. Patients frequently have to switch therapies if their treatment stops being effective, contributing to anxiety. Patients deal with significant side effects.	*“For a lot of diseases and things, you go through a series of treatments, and then you reach a ‘cure’ or things like that. Then you’re able to move on from that, where with the case of myeloma…you don’t reach a finish line.” (White patient, 40s)*
** *Significant financial issues* **	
Medications and treatments are expensive. Several patients reported relying on copay assistance or foundation support to help pay for treatment.	*“I also have a foundation that helps with those prescriptions, because otherwise there’s no way I could pay $27,000 a month for chemo or even my portion of it.”* *(White patient, 60s)*
** *High toll on physical and emotional health* **	
MM has a significant physical and emotional burden. Fatigue takes a major toll. Patients who are not able to engage in common activities can become frustrated and isolated.	*“It’s taken a big toll on my physical and emotional health because…I’m older now and sometimes in my head I have a lot of problems going on.…It is just physically trying to move. Sometimes I’m really slow.” (Black patient, 60s)*
** *Frequent appointments and long stays* **	
Patients reported frequent appointments (e.g., some get their blood tested every 3 weeks). They may also have long hospital stays. Some employers are not willing to accommodate these frequent medical visits and long stays.	*“Sometimes with different cancers or other cancers, they don’t have to go as frequent[ly], say if they’re in remission or what have you. But considering since I am high risk multiple myeloma, I do have more frequent appointments and lengthy stays in the hospital.” (White patient, 50s)*

**Table 3 healthcare-12-01587-t003:** Social needs and identity experiences with negative impact by stage of patient journey (*n* = 19).

Social and Identity Experiences	Diagnosis *n* (%)	Treatment Goals *n* (%)	Treatment Choice *n* (%)	Treatment Initiation *n* (%)	Treatment Adherence *n* (%)	Burden of Disease/Outcomes *n* (%)
Health Knowledge	11 (57.9)	8 (42.1)	10 (52.6)	8 (42.1)	8 (42.1)	8 (42.1)
Financial Instability	5 (26.3)	8 (42.1)	14 (73.7)	7 (36.8)	9 (47.4)	8 (42.1)
Lack of Insurance	7 (36.8)	9 (47.4)	16 (84.2)	11 (57.9)	9 (47.4)	8 (42.1)
Lack of Social Support	7 (36.8)	10 (52.6)	8 (42.1)	8 (42.1)	11 (57.9)	9 (47.4)
Lack of Transportation	3 (15.8)	5 (26.3)	12 (63.2)	9 (47.4)	12 (63.2)	2 (10.5)
Race and Ethnicity	9 (47.4)	7 (36.8)	10 (52.6)	7 (36.8)	5 (26.3)	6 (31.6)
Age	6 (31.6)	11 (57.9)	12 (63.2)	5 (26.3)	7 (36.8)	9 (47.4)
Culture	11 (57.9)	5 (26.3)	9 (47.4)	8 (42.1)	9 (47.4)	6 (31.6)
Disability Status	7 (36.8)	10 (52.6)	14 (73.7)	7 (36.8)	11 (57.9)	14 (73.7)

Responses to question: At which point(s) along the care journey do you think the social and identity experiences can have significant negative impacts? Please select all that apply.

## Data Availability

The original contributions presented in the study are included in the article/Appendix A, further inquiries can be directed to the corresponding author/s.

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
