# Peer review of "Patient Perspectives on Social and Identity Factors Affecting Multiple Myeloma Care: Barriers and Opportunities"

_healthcare, 2024, doi:10.3390/healthcare12161587_

Round 1

Reviewer 1 Report

Comments and Suggestions for Authors

Dear editor and dear authors,

Thank you for the opportunity to review your paper entitled “ Patient Perspectives on Social and Identity Factors Affecting Multiple Myeloma Care: Barriers and Opportunities”

I appreciated the paper; here are some suggestions:

The keywords "patient perspective" and "disease journal" are not MESH descriptors.

Introduction: The authors refer to a previous study about the impact of multiple myeloma in person, but the introduction does not explain why the present study is important.

Materials and methods: Were all the participants in the same disease phase? Please clarify.

Material and methods – "Focus group discussions were audio-recorded and transcribed verbatim". How was the informed consent process?

Do the authors use any data analysis program for data analysis?

The author identifies the study's limitation related to the participant's choice. We suggested improving the conclusions. I recommend that the authors explain to readers the significant impact of these results in clinical practice, as this will help highlight the practical implications of their study.

The references are from the last five years.

Author Response

For research article

Title: Patient Perspectives on Social and Identity Factors Affecting Multiple Myeloma Care: Barriers and Opportunities

Response to Reviewer 1 Comments

1. Summary

2. Questions for General Evaluation

Reviewer’s Evaluation

Response and Revisions

Does the introduction provide sufficient background and include all relevant references?

Must be improved

Please refer to Specific Comment #2 where we provided further clarification on the purpose of the current study.

Are all the cited references relevant to the research?

Yes

Is the research design appropriate?

Yes

Are the methods adequately described?

Yes

Are the results clearly presented?

Yes

Are the conclusions supported by the results?

Yes

3. Point-by-point response to Comments and Suggestions for Authors

** Page numbers correspond to the red-lined version of the Manuscript.

Comment 1: The keywords "patient perspective" and "disease journal" are not MESH descriptors.

Response 1: The authors thank the reviewer for bringing this to our attention. We have more appropriately chosen the keywords “multiple myeloma; patient-centered care; health equity; access to care; perceived social supports” to reflect our scientific work and key findings stemming from this study.

Comment 2: Introduction: The authors refer to a previous study about the impact of multiple myeloma in person, but the introduction does not explain why the present study is important. (page 1)

Response 2: The authors thank the reviewer for allowing us to articulate the importance of this study more clearly. We have made significant changes to the Introduction Section to better describe the results of our previous study while also sharing the significance of this study. “In our previous research using mixed methods, 65% of patients reported ≥1 social need(s) that negatively impacted health outcomes including lack of knowledge about MM, financial instability and lack of insurance, transportation concerns, and lack of social support. [11]. Learning directly from patients where prevalent social needs and experiences occur on the MM diagnostic and treatment is a critical gap that needs to be investigated in order to effectively design mitigation solutions and alleviate MM disease burdens. To build upon the findings of our previous research, we designed a series of exploratory focus groups to identify patients’ priorities with respect to their specific social and identity-related needs. We aimed to map these prioritized needs and experiences to the MM diagnostic and treatment journey, and describe patient-generated ideas to improve the support that patients need. Herein, we report the findings of this follow-up research.” (page 3)

Comment 3: Materials and methods: Were all the participants in the same disease phase? Please clarify.

Response 3: The authors thank the reviewer for the comment and allowing us to clarify in the Materials and Methods section and the Limitation section that participants were all diagnosed with Multiple Myeloma and had received some treatment including (for some) a stem cell transplant. Data on disease phase is difficult to define given the heterogeneity and complexity of Multiple Myeloma disease management. The authors have added to the Limitations section, “Third, while participants were all diagnosed with MM and have received treatment, variation in treatment duration, treatment regimen types and disease phase is expected given the heterogeneity and treatment complexity of the disease”. (page 11)

Comment 4: Material and methods – "Focus group discussions were audio-recorded and transcribed verbatim". How was the informed consent process?

Response 4: We thank the reviewer for your interest in the protection of Human Subjects Research. This is important to our team as well. Please know that study participants were existing members of the Patient Engagement Research Council (PERC). Before any research activities began, participants were provided the consent form virtually and provided ample time to ask questions. They were instructed that no treatments would be provided and they could withdraw at any time. We have revised Section 2.2 Participants to describe the process and approvals in more detail. “At the time of this study, the MM PERC consisted of 23 patients living with MM, who had all have received some treatment, including stem cell transplant. Participants were informed that no treatments would be provided, and they could withdraw at any time. Additionally, a consent and release form were signed by the participants that communicated confidentiality and Health Insurance Portability and Accountability Act (HIPAA)-compliant practices. All data were deidentified and collected for improvement purposes; and an ethics board deemed the study exempt from IRB review according to the Department of Health and Human Services Policy for Protection of Human Research Subjects (Section 45C.F.R46.104). The study was also conducted in accordance with the Helsinki Declaration of 1964 and its later amendments. All sessions were conducted virtually, and participants joined from their homes[12]”. (page 2)

Comment 5: Do the authors use any data analysis program for data analysis?

Response 5: Thank you for the question. For study data analysis, we did not use a data analysis program. Rather, we used a traditional approach where each transcript was manually reviewed, themes were coded and organized according to key research questions and data was reviewed in totality. We thank the reviewer for asking the question and have amended Section 2.4 Data Analysis to reflect the methods that were used with greater transparency. “Transcripts were analyzed using narrative thematic analysis to identify and organize themes according to key research questions related to patient experiences, how social needs and identity experiences map to the patient journey, and key ideas for patient support. Data were reviewed in totality and findings were assessed thematically using qualitative coding to ensure rigor based on the key questions in the discussion guides (Supplemental Material).” (page 2-3)

Comment 6: The author identifies the study's limitation related to the participant's choice. We suggested improving the conclusions. I recommend that the authors explain to readers the significant impact of these results in clinical practice, as this will help highlight the practical implications of their study.

Response 6: We thank the reviewer for the recommendation and have revised the Conclusion section to reflect, “In this study, participants prioritized critical social needs delineated by the disease journey stage, and presented ideas that could be used to improve patient care. Findings of this study will help patient navigators, advocacy groups, community care organizations, as well as health care professionals better support patients with MM as they go through their unique and challenging disease journeys.” (page 11)

Comment 7: The references are from the last five years.

Response 7: Thank you for this observation and commentary. We believe that all citations are recent and relevant to study to explain its significance and help compare current results to existing literature.

Reviewer 2 Report

Comments and Suggestions for Authors

The introduction needs supplementation. Please provide specific results from previous studies on the social determinants and identity-related factors of MM. Following this, state the significance and purpose of this study in overcoming the limitations of these previous studies.

It is recommended to explain the contents of reference [11] if necessary.

In the study design section on page 2, it is mentioned that the groups were formed based on previous research results, but it is difficult to understand the basis for this grouping.

2.2 Please add information regarding IRB approval. Clearly describe the inclusion criteria (diagnosis, disease duration, etc.) and exclusion criteria (presence of comorbid conditions, ability to communicate, etc.) for participant selection.

2.4 Qualitative research, such as focus group interviews, requires rigor in its analysis. Although the appendix explains the procedures for conducting the interviews, it does not sufficiently cover the data analysis process. Please supplement with information on the researchers' preparation for data analysis and efforts to reduce bias.

The results are divided into sections 3.2 and 3.3, but as a reader, I cannot clearly discern the difference between them. It seems necessary to either distinctly differentiate the content or adopt a strategy to integrate them.

Although the text states that Figure 1 summarizes the content in section 3.4, a brief explanation of the figure's content is also needed in the text. Alternatively, guide the reader on the criteria by which the subsequent sections (3.4.1~3.4.7) are organized.

The font in section 3.4.2 is different. Please check the submission format.

This is an important study focusing on patients who rarely but significantly struggle through the disease process. I would appreciate it if you could consider the suggestions provided.

Author Response

For research article

Title: Patient Perspectives on Social and Identity Factors Affecting Multiple Myeloma Care: Barriers and Opportunities

Response to Reviewer 2 Comments

1. Summary

2. Questions for General Evaluation

Reviewer’s Evaluation

Response and Revisions

Does the introduction provide sufficient background and include all relevant references?

Can be improved

Please refer to Comment #1

Are all the cited references relevant to the research?

Yes

N/A

Is the research design appropriate?

Can be improved

Please refer to Comment #2, 3

Are the methods adequately described?

Can be improved

Please refer to Comment #2, 3, and 4

Are the results clearly presented?

Must be improved

Please refer to Comment #5, 6

Are the conclusions supported by the results?

Can be improved

We have revised the conclusion section to reflect, “In this study, participants prioritized critical social needs delineated by the disease journey stage, and presented ideas that could be used to improve patient care. Findings of this study will help patient navigators, advocacy groups, community care organizations, as well as health care professionals better support patients with MM as they go through their unique and challenging disease journeys. (page 11)

3. Point-by-point response to Comments and Suggestions for Authors

Comment 1: The introduction needs supplementation. Please provide specific results from previous studies on the social determinants and identity-related factors of MM. Following this, state the significance and purpose of this study in overcoming the limitations of these previous studies. It is recommended to explain the contents of reference [11] if necessary.

Response 1: The authors thank the reviewer for allowing us to articulate the importance of this study more clearly. We have made significant changes to the Introduction Section to better describe the results of our previous study while also sharing the significance of this current study. In our previous research using mixed methods, 65% of patients reported ≥1 social need(s) that negatively impacted health outcomes including lack of knowledge about MM, financial instability and lack of insurance, transportation concerns, and lack of social support. [11]. Learning directly from patients where prevalent social needs and experiences occur on the MM diagnostic and treatment is a critical gap that needs to be investigated in order to effectively design mitigation solutions and alleviate MM disease burdens. To build upon the findings of our previous research, we designed a series of exploratory focus groups to identify patients’ priorities with respect to their specific social and identity-related needs. We aimed to map these prioritized needs and experiences to the MM diagnostic and treatment journey, and describe patient-generated ideas to improve the support that patients need. Herein, we report the findings of this follow-up research. (page 2)

Comment 2: In the study design section on page 2, it is mentioned that the groups were formed based on previous research results, but it is difficult to understand the basis for this grouping.

Response 2: Thank you for the opportunity to better explain our methodology. The authors have provided further clarity that our research questions for the current study were built off findings from the initial. Section 2.1 Study Design now reflects, “This qualitative study consisted of 2 sets of 4 semi-structured focus group discussions (8 groups in total) in which we built our research questions upon findings from our previous research [11]. The first set of groups focused on understanding patients’ experiences with MM and identifying where along their journeys these factors have the most impact. The second set of groups focused on patient-generated ideas to improve needed support.” (page 2)

Comment 3: Please add information regarding IRB approval. Clearly describe the inclusion criteria (diagnosis, disease duration, etc.) and exclusion criteria (presence of comorbid conditions, ability to communicate, etc.) for participant selection.

Response 3: We thank the reviewer for prompting the study team to include details about ethics board approval. Section 2.2 has been amended. “All data were deidentified and collected for improvement purposes; an ethics board deemed the study exempt from IRB review according to the Department of Health and Human Services Policy for Protection of Human Research Subjects (Section 45C.F.R46.104).” (page 2)

The authors thank the reviewer for the comment and have provided the inclusion/ exclusion criteria for this current study. As previously reported in citation [11],“Participants were adult (≥18 years of age) US residents with MM who were currently in their first or second line of therapy and were existing members of the sponsor’s Multiple Myeloma Patient Engagement Research Council (PERC), which has been described elsewhere [12]. Patients ≤18 years old and those with receipt of stem cell transplant >5 years from time of PERC initiation were excluded.” (page 2)

Comment 4: Qualitative research, such as focus group interviews, requires rigor in its analysis. Although the appendix explains the procedures for conducting the interviews, it does not sufficiently cover the data analysis process. Please supplement with information on the researchers' preparation for data analysis and efforts to reduce bias.

Response 4: Thank you for the question. For study data analysis, we did not use a data analysis program. Rather, we used a traditional approach where each transcript was manually reviewed, themes were coded and organized according to key research questions and data was reviewed in totality. We thank the reviewer for asking the question and have amended Section 2.4 Data Analysis to reflect the methods that were used with greater transparency. “Transcripts were analyzed using narrative thematic analysis to identify and organize themes according to key research questions related to patient experiences, how social needs and identity experiences map to the patient journey, and key ideas for patient support. Data were reviewed in totality and findings were assessed thematically using qualitative coding to ensure rigor based on the key questions in the discussion guides (Supplemental Material).” (page 2-3)

Comment 5: The results are divided into sections 3.2 and 3.3, but as a reader, I cannot clearly discern the difference between them. It seems necessary to either distinctly differentiate the content or adopt a strategy to integrate them.

Response 5: We thank the reviewer for this observation and have made changes to headers describing results in Section 3.2 and 3.3. To better describe the difference in results, the headers now show:

3.2 Describing Key Characteristics of Living With MM

3.3 Comparing MM to Other Chronic Health Conditions

Comment 6: Although the text states that Figure 1 summarizes the content in section 3.4, a brief explanation of the figure's content is also needed in the text. Alternatively, guide the reader on the criteria by which the subsequent sections (3.4.1~3.4.7) are organized.

Response 6. The authors thank the reviewer for the observation that we need to guide the reader more clearly on how our results are organized. We have revised Section 3.4 to include a statement to reflect this. “Subsequent results are organized by disease journey where patients describe the critical social needs at each of the six stages.” (page 5)

Comment 7: The font in section 3.4.2 is different. Please check the submission format.

Response 7: We thank the reviewer for the observation. We have double checked the formatting requirements and will rely on final editorial review by editor to improve formatting if publication is accepted.

Comment 8: This is an important study focusing on patients who rarely but significantly struggle through the disease process. I would appreciate it if you could consider the suggestions provided.

Response 8: We thank the reviewer for their time and contributions. We have made significant improvements based on the comments and questions.
